# Prospective Life Cycle Assessment of Microbial Sophorolipid Fermentation

Karina Balina [1,2,*], Raimonda Soloha [1], Arturs Suleiko [3,4], Konstantins Dubencovs [4,5], Janis Liepins [1] and Elina Dace [1,6]

1   Institute of Microbiology and Biotechnology, University of Latvia, Jelgavas iela 1, LV-1004 Riga, Latvia; janis.liepins@lu.lv (J.L.); elina.dace@lu.lv (E.D.)
2   Institute of Science and Innovative Technologies, Liepaja University, Liela iela 14, LV-3401 Liepaja, Latvia
3   Biotehniskais Centrs AS, Dzerbenes iela 27, LV-1006 Riga, Latvia; arturs.suleiko.00@rpg.lv
4   Laboratory of Bioengineering, Latvian State Institute of Wood Chemistry, Dzerbenes iela 27, LV-1006 Riga, Latvia
5   Institute of General Chemical Engineering, Faculty of Materials Science and Applied Chemistry, Riga Technical University, Paula Valdena iela 3, LV-1658 Riga, Latvia
6   Department of Political Science, Riga Stradins University, Dzirciema iela 16, LV-1007 Riga, Latvia
*   Correspondence: karina.balina@lu.lv

**Abstract:** The biorefinery industry is witnessing a transition from fossil and chemical-based processes to more sustainable practices, with a growing emphasis on using renewable resources. Sophorolipids, a promising group of biosurfactants, present a viable substitute for conventionally produced surfactants. This study focuses on microbial fermentation using yeast and lipid substrate for sophorolipid production. The life cycle assessment (LCA) methodology was employed to identify environmental hotspots of the process and to assess the environmental benefits resulting from the replacement of raw rapeseed cooking oil (base scenario) with waste cooking oil, reduction of process electricity consumption, and increased sophorolipid yield. By compiling scenarios with the lowest environmental impact, a best-case scenario was created. The results revealed that the environmental impact of sophorolipid production could be reduced by 50% in the best-case scenario compared to the base scenario. This research provides valuable insights into the environmental optimization of the fermentation process and through the application of LCA highlights the potential for the reduction of negative environmental impact of sophorolipid production, contributing to the ongoing transition from petroleum oil and petrochemical refineries to sustainable biorefineries.

**Keywords:** glycolipid surfactants; waste cooking oil; environmental impact; life cycle assessment; biotechnology; *Starmerella bombicola*

## 1. Introduction

Biotechnology is viewed as a promising solution to facilitate fossil resource substitution, promote the circular economy, and provide the benefits of renewability and biodegradability [1]. Fermentation is a biotechnological process that uses microorganisms such as bacteria and yeast to produce value-added chemicals, including biosurfactants [2].

Sophorolipids are microbial biosurfactants with versatile structures and properties such as detergency, solubilization, foaming capacity, and lubrication [3]. Previous studies have mainly focused on the use of different types of substrates, including glucose and vegetable oil, for sophorolipid production [4–7]. While sophorolipids are considered to have a lower environmental impact compared to conventional surfactants derived from petrochemicals due to their biodegradability, low toxicity, and renewable origin, the high costs of raw materials and production process have hindered their market entry [8]. However, there is a lack of evidence-based assessment of the environmental sustainability of microbial biosurfactant production processes [9,10].

Only a limited number of studies have investigated the environmental impacts of biosurfactant production [6,7,11]. The environmental impact of sophorolipid production is influenced by factors such as the fermentation substrate used, energy consumption, and generation of waste biomass during the different stages of the production process [9,11]. To evaluate the environmental impact of the production process and identify areas for improvement, life cycle assessment (LCA) can be conducted. LCA is a widely accepted and standardized methodology for evaluating the environmental impacts of a product or process throughout the entirety or a part of its life cycle. LCA studies help to identify the environmental hotspots in the life cycle of a product or process and provide insights into how to improve its environmental performance [12]. Prospective LCA can play a crucial role in the development of biorefinery processes as it allows a systematic evaluation of the environmental impacts and guidance regarding sustainability [13–15].

To evaluate the environmental impact of a product, process, or service that is still in the planning stage, a prospective LCA methodology is applied [16,17]. This methodology helps to identify potential environmental impacts and improvement opportunities early in the development process. The prospective LCA methodology models the life cycle of the system using assumptions and estimations. The results of the model are then used to evaluate the potential environmental impacts and identify areas where improvements can be made [17]. Despite its potential to provide environmental guidance for developers and to assess future environmental impacts, the prospective LCA methodology has not yet been applied to the biosurfactant production process, and there is no information available on this topic in the existing literature. However, it is worth noting that the dynamic LCA approach has been utilized by Hu et al. to analyse sophorolipid production from organic waste streams, identify hotspots, and derive recommendations to reduce negative environmental impacts [18].

One key benefit of the LCA methodology is that it allows identification of areas in the life cycle of a product or process that can be altered to reduce the environmental impact. For example, LCA can identify opportunities for optimized fermentation conditions to reduce electricity consumption during the production process [19,20]. LCA can also be used to quantify the potential environmental impact of using different substrates for biosurfactant production, such as glucose, vegetable oil, or waste cooking oil [6,17]. Substituting rapeseed oil with the waste cooking oil presents an opportunity for circular economy practices by repurposing a resource that would otherwise be discarded, thereby reducing the environmental burden associated with raw material extraction and waste generation [21].

By simulating the use of different substrates, LCA results facilitate the selection of substrate with the lowest environmental impact. Thus, prospective LCA facilitates prediction and avoidance of environmental burden and reduces trial and error costs early in the technology development process. Nevertheless, there is still a need for more LCA studies on lab- and pilot-scale reactors. Such studies provide more accurate and realistic assessments of the environmental impact of biotechnology. Overall, the use of LCA studies can optimize the sustainability of bio-based production processes [22] and minimize the environmental impact of circular bio-based products. The novelty of the prospective LCA methodology lies in its capacity to predict environmental risks during the technology development stage, enabling the avoidance of unnecessary costs and experiments by providing a systematic and data-driven approach to assess and optimize the environmental impact of emerging technologies.

This study focuses on improving the environmental performance of sophorolipid production using raw rapeseed cooking oil (RCO) as a lipid substrate in a lab-scale bioreactor. Prospective LCA methodology was used to quantify the environmental impacts of the process (environmental hotspots) and to identify alternative production scenarios with lower negative impact on the environment. The primary aim of this article is to explore ways to improve the environmental sustainability of sophorolipid fermentation.

## 2. Materials and Methods

*2.1. Life Cycle Assessment*

An environmental impact assessment was conducted using the Life Cycle Assessment (LCA) methodology to quantitatively evaluate the environmental impact of sophorolipid production using vegetable oil as a substrate. This study was carried out in accordance with ISO 14040:2006 and ISO 14044:2006 standards [12,23]. The LCA model was developed in four phases (Figure 1).

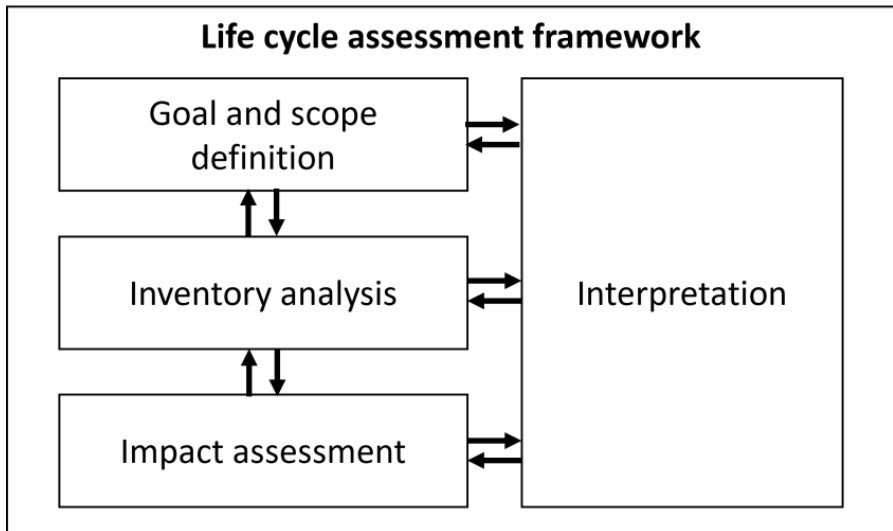

**Figure 1.** LCA phases (adapted from [12,23]).

In the first phase, the study's goal, function, and functional unit were defined, and system's boundaries were set. The second phase involved inventory analysis and accounted for all input/output flows related to the studied system—sophorolipid production. In the third, impact assessment, phase, the inventory data were assessed using key indicators to produce the environmental profile of each scenario. Finally, in the interpretation phase, the results were analysed, and systematic checks of the assumptions and data were conducted to ensure the robustness of results.

To build the inventory, primary data were collected from the lab-scale bioreactor, and secondary data were obtained from literature and the ecoinvent database. The collected data were then used to assess environmental impacts using SimaPro 9.3 software, developed by PRé Sustainability (PRé Sustainability B.V., Amersfoort, The Netherlands). Ecoinvent 3.8 supported the data processing to create the LCA model and evaluate the overall environmental impact using the ReCiPe 2016 method (ecoinvent, Zurich, Switzerland). The ReCiPe Endpoint method (ReCiPe Endpoint (H)/World ReCiPe H/A) was selected to evaluate the environmental impact of the sophorolipid production process. Results are expressed as single score units which are calculated by applying weighting factors to each impact category. This allows for a simplified comparison of the environmental performance of different products or scenarios [24].

The hypothesis for the research is that the use of WCO as a feedstock in the production process will result in a lower overall environmental impact compared to the use of RCO. Additionally, it is hypothesized that energy consumption will have the highest environmental impact, and that a reduction in the total environmental impact of the production process can be achieved by optimizing the fermentation process. Furthermore, it is hypothesized that the combination of the best scenarios will allow for a reduction in the total environmental impact by at least 30%.

The first two phases of the LCA are more methodological and related to data gathering, and thus are described in the next subsections. The last two phases, namely life cycle impact assessment and life cycle interpretation, are discussed in Section 3.

### 2.2. Goal and Scope Definition

The goal of this study was to find the combination of yeast fermentation process conditions for producing sophorolipids from vegetable oil at a lab-scale while minimizing the negative environmental impact of the process. For this, the prospective LCA method was applied. Performing an LCA in the early stage of technology development proves instrumental in mitigating potential environmental degradation. This anticipatory, or prospective, LCA provides results that serve as a comprehensive guide offering valuable suggestions for refining the technology's design and development process when scaling up to industrial size. The study was carried out to improve the environmental impact of the novel bioreactor and serve as a guideline in avoiding negative environmental impacts during the technology's upscaling, as well as to provide recommendations to manufacturers on reducing environmental impact.

The product system assessed in this study includes the following stages of a fermentation process: pre-fermentation, fermentation (biomass stage), fermentation (sophorolipid stage), and filtration. A more detailed description of each process is provided in Section 2.

The scope of this study includes the raw material production system process—rapeseed oil production system, nutrients, and electricity to provide necessary conditions for the yeast biomass. It is a cradle-to-gate assessment, gate being the produced sophorolipid biosurfactant before distributing for further exploitation as shown in Figure 2.

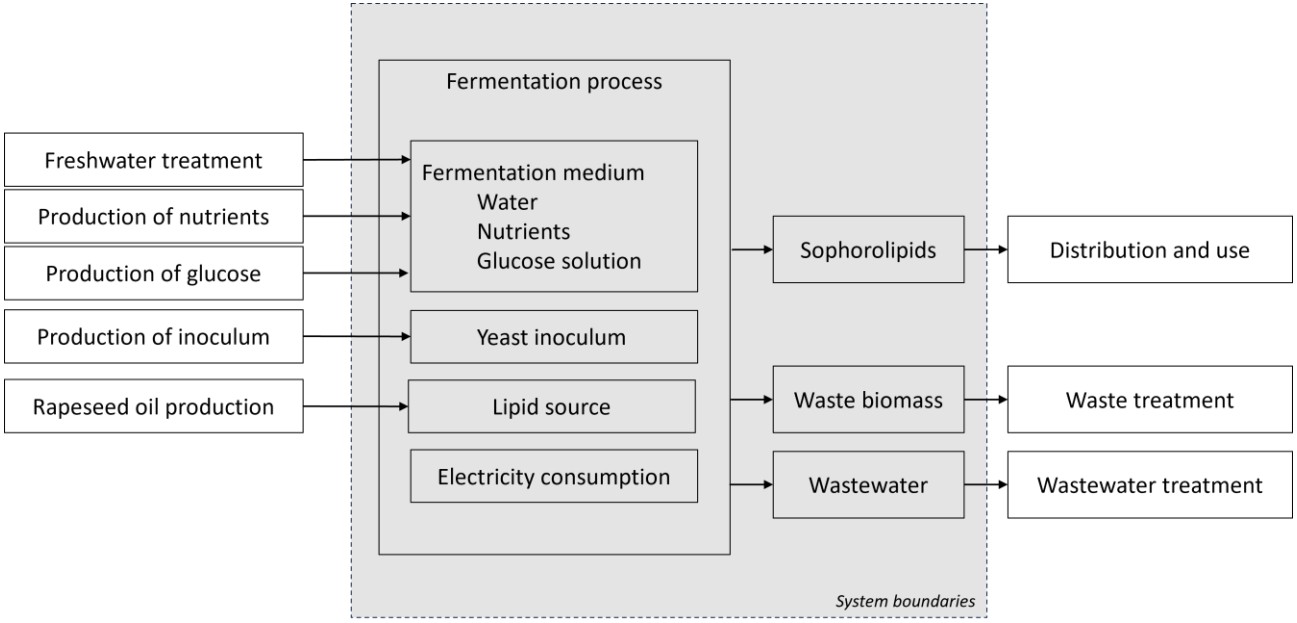

**Figure 2.** Scope and boundaries of the cradle to gate LCA study.

After identification of environmental hotspots, scenarios to reduce the environmental impact were developed and assessed. The goal of scenario development was to identify the highest sophorolipid titre with the least environmental impact.

The functional unit used for this study was the production of 1 kg of sophorolipids from vegetable oil in a 5 L lab-scale bioreactor, using the yeast *Starmerella bombicola*. It is crucial to emphasize that at the stage of the research, the field of sophorolipid application was not yet explicit. Consequently, 1 kg sophorolipid production was used as a reference unit.

The system boundary was defined by selecting the process units that are going to be included in the investigated system: raw material resources, purchased nutrients and reagents used in the fermentation medium and the fermentation process itself (Figure 2). Background data of all product systems include extraction and production processes of all inputs and any additional activities required to make each process functionally equivalent, such as the manufacturing of inputs or the heat and electricity generation. It also includes the effects of co-products along the supply chain.

Sophorolipid distribution, application, and end-of-life treatment were not included in this study. Infrastructure elements, such as the construction of the plant, were not included, as the final bioreactor assembly was not reached at the stage of performing the LCA study.

In this process, aerobically produced yeast is a by-product of the fermentation process and can be reused rather than purchased and stocked as dormant yeast. This avoids the need to produce inoculum to bring dormant yeast to optimal conditions. However, recirculation of any products or co-products was not included in the LCA study to avoid allocation problems, hence they were studied as waste flows.

The system boundary allowed the exclusion of any flows predicted to contribute less than 1% of any impact category from the inventory. A cut-off benchmark criterion of 1% for mass or energy flows was established to ensure that no more than 5% of total flows were excluded from the study, since they wouldn't significantly change the overall conclusions of the study. For this reason, minor components of nutrient medium, stabilisers and reagents used for measurements were excluded from the study.

Upstream allocation, which excludes the environmental impact from any upstream processes, was applied in the scenario, where waste cooking oil (WCO) was used as an input material [25]. In scenarios, WCO has the same function as RCO, thus substituting it completely. WCO is accounted for credits for the avoided production of RCO [26].

Foreground data pertains to the materials and energy used in the fermentation process, while background data relates to the consumed resources and environmental impacts associated with the production process of those materials and energy. In most instances, the readily accessible foreground data within the database were used. However, to secure the necessary background data, information pertaining to the cultivation of rapeseed and its subsequent oil extraction under Latvian conditions was utilized and included as the product system. This data was extracted from the LCA study by Fridrihsone (2020), which examined the production of polyol monomers using rapeseed oil [27].

Primary data to support the LCA study were gathered from a lab-scale experimental bioreactor (base scenario). Secondary data on electricity and production of nutrients for the feeding medium were obtained from the ecoinvent 3.8 database. Furthermore, assumptions based on scientific literature and own data calculations were utilized to support primary data.

### 2.3. Life Cycle Inventory Analysis

Life cycle inventory (LCI) accounts for all inputs and outputs related to the sophorolipid production system.

All input and output data of the sophorolipid production system were collected and summarized in Table 1 and are described in the following subsections.

**Table 1.** Life cycle inventory for sophorolipid production process.

| | Key Parameter | Unit | Value | Data Source |
|---|---|---|---|---|
| **Pre-fermentation** | INPUT | | | |
| | Yeast inoculum | g | 100 | New process—Yeast inoculum |
| | Fermentation medium | g | 1900 | New process—Fermentation medium |
| | Length of process | h | 24 | |
| | Power usage | kW | 0.16 | ecoinvent v3.8, Medium voltage, LV electricity mix |
| | OUTPUT | | | |
| | Pre-fermented substrate | kg | 2 | |
| **Biomass growth stage (A)** | INPUT | | | |
| | Pre-fermented substrate | kg | 2 | |
| | Fermentation medium | kg | 1 | New process—Fermentation medium |
| | Length of process | h | 48 | |
| | Power usage | kW | 0.3 | ecoinvent v3.8, Medium voltage, LV electricity mix |
| | OUTPUT | | | |
| | Fermented biomass A | kg | 2.1 | |
| **Sophorolipid production stage (B)** | INPUT | | | |
| | Fermented biomass A | kg | 2.1 | |
| | Rapeseed oil (RCO) | kg | 0.4 | New process—Rapeseed oil [23] |
| | Glucose solution | kg | 0.24 | New process—Glucose solution |
| | Length of process | h | 161 | |
| | Power usage | kW | 0.42 | ecoinvent v3.8, Medium voltage, LV electricity mix |
| | OUTPUT | | | |
| | Fermented substrate B | kg | 2.74 | |
| **Downstream processing** | INPUT | | | |
| | Fermented substrate B | kg | 2.74 | |
| | Electricity consumption | kWh | 0.3 | ecoinvent v3.8, Medium voltage, LV electricity mix |
| | OUTPUT | | | |
| | Sophorolipids | g | 537.9 | |
| | Water + salt residuals | kg | 2.17 | ecoinvent v3.8, Wastewater from ADof whey {GLO} |
| | Biomass | g | 32.3 | |

### 2.3.1. Yeast Strain and Medium

The *Starmerella bombicola* DSM 27465 strain was obtained from the German Collection of Microorganisms and Cell Cultures (DSMZ, Leibniz Germany). The *S. bombicola* was maintained on YPD agarised media. The YPD medium contained (grams per litre) 10 g of yeast extract (Biolife, Milano, Italy), 20 g of bactopeptone (Biolife, Milano, Italy), 20 g of glucose (Sigma, Taufkirchen, Germany), and 20 g of agar (Biolife, Italy). This step is not included in inventory analysis as a separate product unit.

The inoculum for lab-scale bioreactor fermentations was prepared in 100 mL shake flasks and maintained for 24 h at 30 °C and agitated at 180 rpm in an ES-20 orbital shaker-incubator (Biosan Ltd., Riga, Latvia). The fermentation medium for biomass growth (inoculum preparation and laboratory bioreactor cultivations) and sophorolipid production phases were adapted from Kim et al. [28] and are provided in Table 2.

The process of water distillation was not available in the database. It was assumed that water used in the process was distilled using a water distiller with average energy consumption of 0.7 kWh $L^{-1}$.

**Table 2.** Inventory associated with the production of 1 L fermentation medium.

| Data | Unit | Value | Data Source |
|------|------|-------|-------------|
| **Input** | | | |
| Yeast (inoculum) | g | 10 | Not included in study |
| Glucose | g | 30 | ecoinvent v3.8, Glucose {GLO} |
| Yeast extract | g | 5 | Not included in study |
| $KH_2PO_4$ | g | 1 | ecoinvent v3.8, Assumed as Chemical, inorganic {GLO} |
| $MgSO_4 \times 7H_2O$ | g | 0.5 | ecoinvent v3.8, Assumed as Magnesium sulfate {GLO} |
| $CaCl_2 \times 2H_2O$ | g | 0.1 | ecoinvent v3.8, Assumed as Calcium chloride {GLO} |
| NaCl | g | 0.1 | ecoinvent v3.8, Sodium chloride, powder {GLO} |
| Meat peptone | g | 0.7 | Not included in study |
| Distilled $H_2O$ | g | 952.6 | New process—Distilled water |
| Electricity | kWh | 28.8 | ecoinvent v3.8, Medium voltage, LV electricity mix |
| **Output** | | | |
| Inoculum | g | 1000 | Assumed that 1 L = 1000 g |

### 2.3.2. Lab-Scale Fed-Batch Fermentation

Lab-scale fed-batch fermentations were performed in a 5.4 L working volume bioreactor EDF-5.4_1 JSC Biotehniskais centrs (Riga, Latvia). The fermentation was run in two stages: (1) biomass growth stage and (2) sophorolipid production stage (Figure 3).

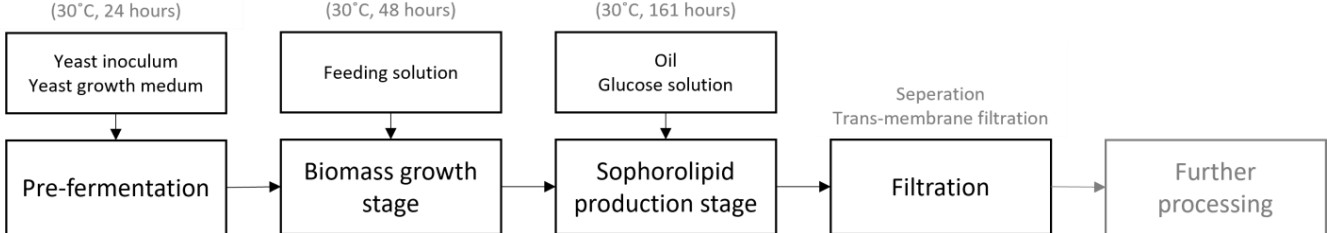

**Figure 3.** Sophorolipid production stages.

During the biomass growth stage, the temperature, pH, and dissolved oxygen (DO) were set to 30.0 °C, 5.0 ± 0.2, and 30 ± 5%-sat., respectively. Active control of the process parameters and foam level was implemented using the BIO-4 bioprocess controller JSC Biotehniskais centrs (Riga, Latvia). The biomass growth stage took place for 48 h after introducing the seed material into the bioreactor. The sophorolipid production stage was initiated by introducing RCO (50 g per litre) and glucose (30 g per litre) into the fermentation medium, while the temperature and pH setpoint values were changed to 25 °C and 3.5, respectively. Feeding of RCO was maintained daily—by adding 20 g of oil per Liter of fermentation medium. RCO was purchased from a local supermarket. Inventory data for locally produced RCO were adapted from the LCA study reported by Fridrihsone and included in the study as a product system [27]. The inventory unit process of rapeseed and rapeseed oil production is summarized in Table S1. The glucose feeding (using an aqueous solution of 400 g $L^{-1}$ glucose (Sigma, Taufkirchen, Germany)) was done continuously to maintain a glucose concentration of about 30 g per litre in the fermentation medium. The inventory of glucose solution is documented in Table 3.

The pH control of the fermentation medium was performed by automatic dosing of acidic (20 W% $H_2SO_4$) or alkali (6 N NaOH) solutions to the fermentation medium via two pre-calibrated peristaltic pumps. The foam level was maintained by automatic dosing of an antifoam agent (Antifoam 204 Sigma, Taufkirchen, Germany) into the fermentation medium via a pre-calibrated peristaltic pump. The DO control was performed according to a cascade algorithm (primarily increasing the agitation rate from 100 to 1000 rpm, and, secondly, by enriching the inlet gas with oxygen). During the cultivations, a constant gas flow rate of 1.6 slpm (standard Liters per minute) was maintained.

**Table 3.** Inventory associated with the production of the glucose solution.

| Data | Unit | Value | Data Source |
| --- | --- | --- | --- |
| Input | | | |
| Glucose | g | 400 | ecoinvent v3.8, Glucose {GLO} | |
| Distilled water | g | 560 | New process—Distilled water |
| Output | | | |
| Glucose solution | g | 1000 | Assumed that 1 L = 1000 g |

Process data acquisition was performed through a SCADA (supervisory control and data acquisition) software which was linked to a BIO4 bioprocess controller.

### 2.3.3. Quantitative Measurements of Biomass, Sophorolipids, and Residual Oil

Off-line samples for biomass, sophorolipid, and residual oil concentration measurements were harvested aseptically every 24 h. Medium samples were centrifuged at $1327 \times g$ for 15 min and stored at $-21\,°C$ for further analyses.

The biomass concentration was determined only during the growth stage by measuring the optical density at 600 nm with a Jenway 6300 (Staffordshire, UK) spectrophotometer. The biomass dry cell weight (DCW) in relation to the absorbance was determined gravimetrically by drying the medium sample and weighing the leftover biomass. The correlation coefficient between biomass concentration per litre and optical absorbance was determined during the exponential growth phase and was equal to $0.13$ g DCW $L^{-1}$ $A.U^{-1}$.

The residual oil was measured by extraction with n-hexane. After phase separation, the top layer containing the oil was separated and evaporated using a rotary vacuum evaporator Stuart RE400 (Keison Ltd., Essex, UK) and then weighed.

After the extraction of residual oil, the sample was extracted three times with ethyl acetate (50 mL of ethyl acetate to 10 mL of sample each). Next, the ethyl acetate layer was separated and evaporated using a rotary vacuum evaporator Stuart RE400 (Keison Ltd., UK) and weighed.

### 2.3.4. Separation of Biomass and Sophorolipids

After the end of the sophorolipid production phase, the fermentation medium was harvested and stored at $2\,°C$ for further sophorolipid separation. The separation was done using a crossflow filtration apparatus (Figure 4).

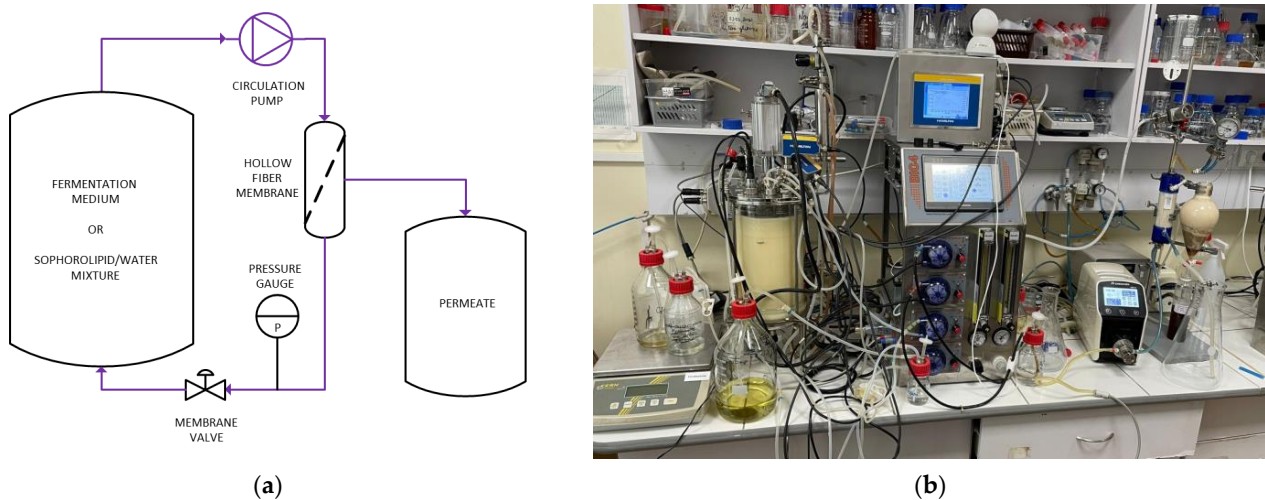

(**a**)　　　　　　　　　　　　　　　　　　　　　　　　　　　　　(**b**)

**Figure 4.** (**a**) Schematic drawing of the biomass separation and sophorolipid concentration apparatus, (**b**) photo of the lab-scale bioreactor.

The downstream processing was divided into two sub-stages, i.e., (1) biomass separation stage and (2) sophorolipid concentration stage.

In the biomass separation stage, yeast cells and other debris were separated from the fermentation medium. For biomass separation, a PMMA ZENA P6 membrane with pore size of 0.2 µm and filtration surface of 5 m$^2$ (ZENA Membranes, Czech Republic) was used. The trans-membrane pressure was maintained at 1 bar, while the retentate flow was adjusted to 500 mL min$^{-1}$. The biomass separation procedure was run until at least 1 L of sophorolipid/fermentation medium mixture (permeate) was generated or in other cases for 1 h.

After biomass separation, the permeate was further used for sophorolipid concentration. For sophorolipid concentration, a PMMA membrane Toray BK-2.1P (Toray Inc., Santa Ana, CA, USA) (M.W.C.O.—25 kDa, filtration surface—2.1 m$^2$) was used. The trans-membrane pressure was maintained at 0.1 bar, while the retentate flow was adjusted to 500 mL min$^{-1}$. After generating at least 0.5 L of concentrated sophorolipid mixture or in other cases for 1 h, the sophorolipid and residual oil was quantified using the measurement methods described above.

### 2.4. Scenario Development

In this study, scenario development was employed to assess the potential environmental benefits of various alternatives in the microbial sophorolipid fermentation process. The process involves the development and evaluation of alternative scenarios to analyse the potential effects of different fermentation conditions and to understand their impact on the environmental performance of the process.

Key parameters that significantly influence the environmental performance of the sophorolipid production process were identified in the life cycle impact assessment phase and defined as environmental hotspots. The base scenario serves as the reference point for comparison with alternative scenarios. The utilization of RCO as the substrate for sophorolipid production was used as the base scenario. Subsequently, alternative scenarios were formulated, focusing on two key factors, established in hotspot analysis: the replacement of raw cooking oil with waste cooking oil and the reduction of electricity consumption during the fermentation process, which also includes efficient use of the reactor chamber.

## 3. Life Cycle Impact Assessment and Interpretation

The life cycle impact assessment (LCIA) phase links inventory flows to selected indicators for the LCA. This process involves determining the relationship between each flow and its corresponding impact indicator, followed by the selection of an appropriate characterization model that quantifies the relationship between each inventory type and its related indicator. For example, emissions of carbon dioxide and methane are both known to contribute to the climate change indicator.

### 3.1. Environmental Hotspot Analysis

The environmental hotspots were identified through the application of the LCIA methodology. The ReCiPe Endpoint methodology was utilized to evaluate the process contributions in three damage categories: resources, ecosystems, and human health. Results were aggregated after normalization and weighting for comparison purposes. The base scenario used in this study was sophorolipid production with RCO as a substrate.

The aggregated results of the base scenario indicate that the use of RCO as a substrate contributes to 14.2% of the total environmental impact (1754 mPt single score units) of sophorolipid production process (Figure 5). Similar conclusions were made by Baccile et al. [7] who conducted a comprehensive study on acetylated acidic sophorolipids, indicating that the production phase, particularly from substrates like glucose and rapeseed oil, has the highest impacts. They emphasize the importance of optimizing the substrate ratio and using second-generation raw materials for improved sustainability [7]. The largest negative environmental impact, i.e., 85.1%, is generated by electricity that is used to provide

energy for the sophorolipid production processes. Aru and Ikechukwu's gate-to-gate LCA of biosurfactants also highlighted the significant contribution of emissions from electricity supply to overall environmental impacts [29]. A negligible, 0.4%, environmental impact was caused by the glucose used as a carbohydrate source.

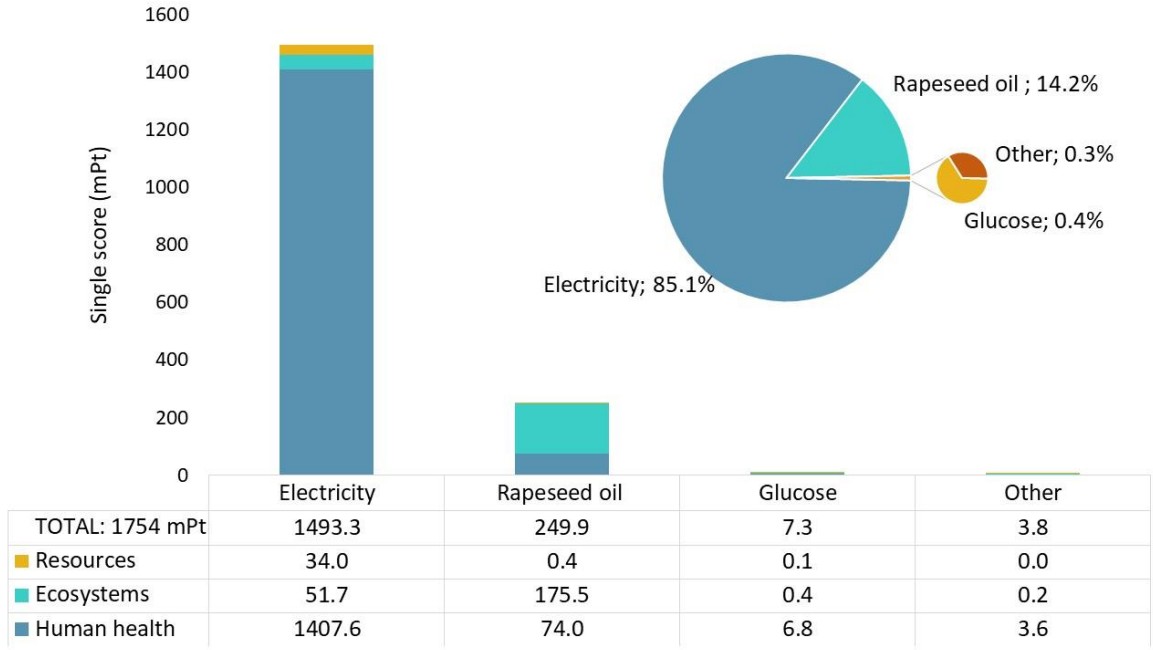

| TOTAL: 1754 mPt | Electricity | Rapeseed oil | Glucose | Other |
|---|---|---|---|---|
| | 1493.3 | 249.9 | 7.3 | 3.8 |
| Resources | 34.0 | 0.4 | 0.1 | 0.0 |
| Ecosystems | 51.7 | 175.5 | 0.4 | 0.2 |
| Human health | 1407.6 | 74.0 | 6.8 | 3.6 |

**Figure 5.** Top contributing elementary flows in the sophorolipid production process base scenario.

In terms of damage categories, the LCIA results indicate that ecosystems receive most of the damage from RCO usage, while electricity has a significant impact on human health. Electricity causes impacts on human health due to the release of particulate matter (PM) during the combustion of fossil fuels and solid biofuel in power plants. PM emissions contribute to air pollution and can cause respiratory and cardiovascular issues, making it crucial to consider these health effects in LCIA studies [30]. Meanwhile, rapeseed oil production impacts ecosystems through land use change, agricultural practices, and potential chemical inputs, affecting biodiversity, soil health, and water resources. The environmental impact of the production process can be reduced through optimization of the process. This can involve selecting a different substrate, improving technologies, and adjusting growth parameters to reduce electricity consumption.

### 3.2. Solutions to Reduce Environmental Impact

To reduce the negative environmental impact of the sophorolipid production process, two process optimization scenarios were developed.

The first scenario (S1) aimed to assess the environmental impact of sophorolipid fermentation by replacing the growth substrate and changing the sophorolipid titre Table 4.

**Table 4.** Variations of the first scenario in comparison to the base scenario.

| Scenario | S-0 | S1-B | S1-C | S1-D | S1-E | S1-F | S1-G | S1-H | S1-I |
|---|---|---|---|---|---|---|---|---|---|
| Type of substrate | RCO | WCO | WCO | WCO | WCO | WCO | WCO | WCO | WCO |
| Sophorolipid titre compared to the titre in base scenario | Base scenario | −30% | −20% | −10% | 0% | 10% | 20% | 30% | 50% |
| SL titre (g L$^{-1}$) | 196.3 | 137.4 | 157.1 | 176.7 | 196.3 | 215.9 | 235.6 | 255.2 | 294.5 |

Environmental hotspot analysis of the base scenario (S-0) indicated that the use of RCO is the second largest contributor to the environmental impact. This impact is mainly attributed to agricultural activities associated with the cultivation of rapeseed, including the use of fertilizers [27]. While the environmental impact of other vegetable oils may differ, it is likely to remain similar to rapeseed oil [31]. Thus, in S1, RCO was replaced by waste cooking oil (WCO), i.e., oil that has been exposed to high heat in the frying process [3] and is not usable in food anymore. Utilizing WCO as a lipid source in the sophorolipid fermentation process can help avoiding the environmental impact caused by RCO. In the life cycle of WCO, the environmental impact of its production is allocated to its former life cycle that ends with the cooking process. Thus, the environmental impacts associated with rapeseed cultivation are avoided, i.e., assumed to be zero.

The obtained aggregated normalized and weighed ReCiPe endpoint results indicate that a 28% reduction in environmental impact is achieved, assuming that the sophorolipid outcome is the same as for the base scenario (S1-E in Figure 6). However, it should be noted that when developing LCA scenarios, results from shake flask cultivations with raw and WCO were included. In shake flask cultivations reported by Liepins et al. [32], rapeseed WCO yielded 50% higher sophorolipid titres than RCO. According to their laboratory experiments, the higher sophorolipid outcome can be achieved without violating mass conservation conditions. The study by Kaur et al. suggests that during the cooking process of RCO, triple and double moieties are damaged [33]. Fatty acids with a single moiety are more readily available to be transformed into biosurfactants [34]. Based on the mass balance calculation of sophorolipid production, a 50% higher titre was assumed in scenario S1-I compared to the base scenario, deriving a decrease in negative environmental impact by 52% (S1-I in Figure 6). To simulate a less optimistic scenario, a 30% lower sophorolipid production outcome was tested (S1-B in Figure 6). The LCIA results reveal that this would lead to a 3% increase in the environmental impacts. Even if the sophorolipid production process using WCO would give a 20% lower titre (S1-C in Figure 6), the environmental impact would decrease by 10% compared to the base scenario.

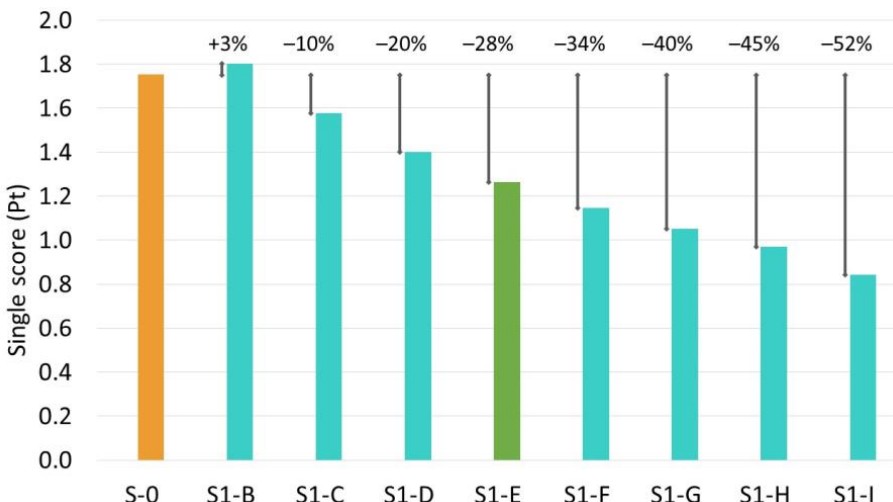

**Figure 6.** Environmental impact results of the first scenario (RCO replacement by WCO).

The second scenario (S2) involves energy-saving measures, reducing electricity consumption (kWh) per 1 kg of produced sophorolipids. Various scenarios were formulated by analysing several variations of fermentation parameters that could be altered, aiming to assess the potential changes in environmental impact and propose reduction strategies (Table 5)

**Table 5.** Variations of the second scenario in comparison to the base scenario. Arrows indicate the value in comparison with the base scenario.

| Scenario | S-0 | S2-B | S2-C | S2-D | S2-E | S2-F | S2-G |
|---|---|---|---|---|---|---|---|
| **Biomass growth stage** | | | | | | | |
| Glucose solution (kg) | 0 | 1.14↑ | 1.14↑ | N/A | 0 | 0 | 0 |
| Length of the process (h) | 48 | 138↑ | 138↑ | N/A | 48 | 48 | 48 |
| Power usage (kW) | 0.3 | 0.3 | 0.3 | N/A | 0.3 | 0.3 | 0.3 |
| **Sophorolipid production stage** | | | | | | | |
| Glucose solution (kg) | 0.24 | 0.36↑ | 0.36↑ | N/A | 0.24 | 0.24 | 0.24 |
| Oil (kg) | 0.4 | 0.36↓ | 0.36↓ | N/A | 0.4 | 0.4 | 0.4 |
| Length of the process (h) | 161 | 130↓ | 130↓ | N/A | 161 | 161 | 161 |
| Power usage (kW) | 0.42 | 0.42 | 0.42 | N/A | 0.42 | 0.42 | 0.42 |
| Electricity consumption compared to base scenario (%) | 0% | N/A | N/A | +10%↑ | −5%↓ | −10%↓ | −15%↓ |
| Total volume of the fermentation substrate (L) | 2.74 | N/A | N/A | 3.99↑ | N/A | N/A | N/A |
| SL titre (g L$^{-1}$) | 196.3 | 83.2↓ | 196.1 | 196.1 | 196.1 | 196.1 | 196.1 |

In scenarios S2-B and S2-C, the electricity consumption was reduced by decreasing the length of the sophorolipid fermentation process, while in the scenario S2-D, the maximum working volume of the lab-scale bioreactor chamber was utilized.

The fermentation time in S2-B and S2-C was decreased by stimulating the biomass growth with an added glucose in the biomass growth stage. To balance the biomass loss caused by shorter fermentation time, extra glucose feeding was used in the biomass cultivation stage. This scenario was tested in the lab-scale bioreactor, and this batch yielded a lower sophorolipid titre, but a higher amount of the biomass. Figure 7. Environmental impact results for the second scenario (S-0—base scenario, S2-B—decreased fermentation time, increased titre, S2-C—decreased fermentation time, same titre as base scenario, S2-D—optimised bioreactor working volume)shows that this approach resulted in a 150% higher environmental impact than the base scenario. The higher environmental impact in scenario S2-B can be attributed to the extended length of the biomass growth stage, despite the reduction in sophorolipid fermentation time. Additionally, the lower sophorolipid titre obtained in the experiment contributed to the higher environmental impact in this scenario.

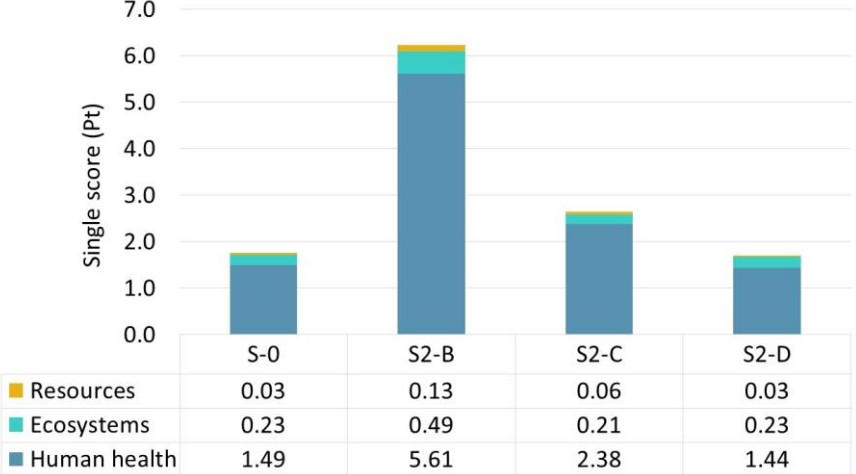

**Figure 7.** Environmental impact results for the second scenario (S-0—base scenario, S2-B—decreased fermentation time, increased titre, S2-C—decreased fermentation time, same titre as base scenario, S2-D—optimised bioreactor working volume).

In scenario S2-C, it was assumed that the sophorolipid titre would be the same as in the base scenario (S-0). However, this scenario exhibited a 50% increase in environmental impact compared to the base scenario.

In scenario S2-D, the working volume of the bioreactor was increased from 2.7 L to 4 L within a 5 L bioreactor. Furthermore, it was assumed that the sophorolipid titre would remain constant, equivalent to that of the base scenario. Optimisation of the working volume of the bioreactor displayed the environmental impact reduction. It was assumed that the increase of working volume would increase the electricity consumption by 10%. Full utilization of the bioreactor chamber made it possible to reduce the environmental impact by nearly 5%.

Scenarios S2-E, S2-F, and S2-G assume an increase in energy efficiency by 5%, 10%, and 15% achieved by lower temperature, reduced mixing speed or reduced heat loss from the process. These are realistic energy-saving options when scaling up the reactor size. It was assumed that the process would yield the same amount of sophorolipids as in the base scenario.

The results in Figure 8 show that 5% electricity savings would reduce the environmental impact by 4%; 10% electricity savings would reduce the environmental impact by 7%; and 15% savings would reduce the impact by 11%, as compared to the base scenario.

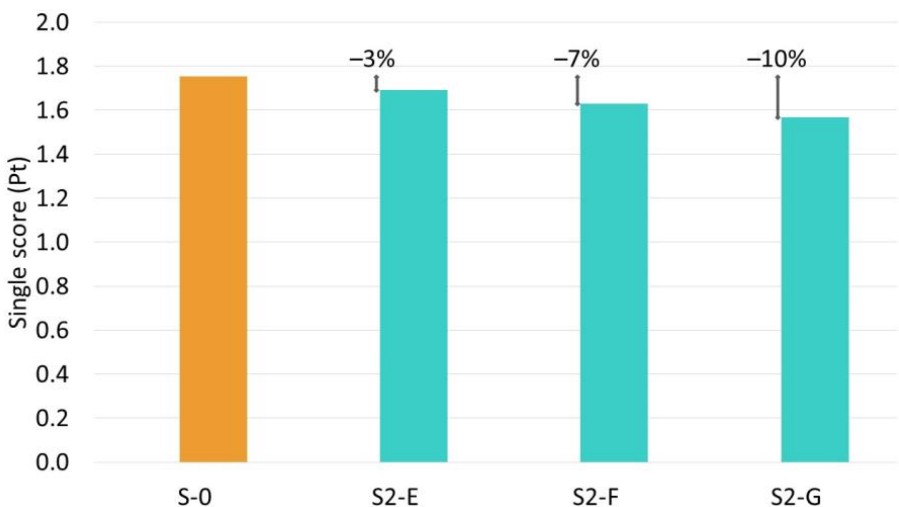

**Figure 8.** Environmental impact results for the second scenario (S-0—no electricity savings, S2-E electricity savings by 5%, S2-F—by 10%, S2-G—by 15%).

A compilation of the most environmentally favourable scenarios was devised to establish a best-case scenario for assessing the potential maximum reduction in environmental impact during sophorolipid production. Specifically, Scenario S1-I, which achieved a 50% higher titre using WCO, was combined with Scenario S2-D, involving optimized bioreactor working volume along with a 10% increase in electricity consumption, and Scenario S2-G, which proposed a 15% reduction in electricity consumption, resulting in a cumulative improvement of 5%. A combination of the results of the simulated scenarios S1-I, S2-D, S2-G would potentially lead to a 60% reduction of the environmental impact compared to the base scenario (Figure 9).

Combining the best-performing scenarios resulted in reduced environmental impact in several impact categories. The main impacts of the studied base scenario are attributed to fine particulate matter (PM) formation, global warming, and land use. As shown in Figure 9, the combined scenario would make it possible to reduce the environmental impact in all impact categories, land use being the most significant. By optimizing the production process, not only can the overall environmental burden be reduced, but it also becomes possible to avoid adverse effects on land use and ecosystems. This positive outcome is achieved by utilizing WCO as a substrate for sophorolipid production, thereby eliminating the need for rapeseed cultivation and RCO production.

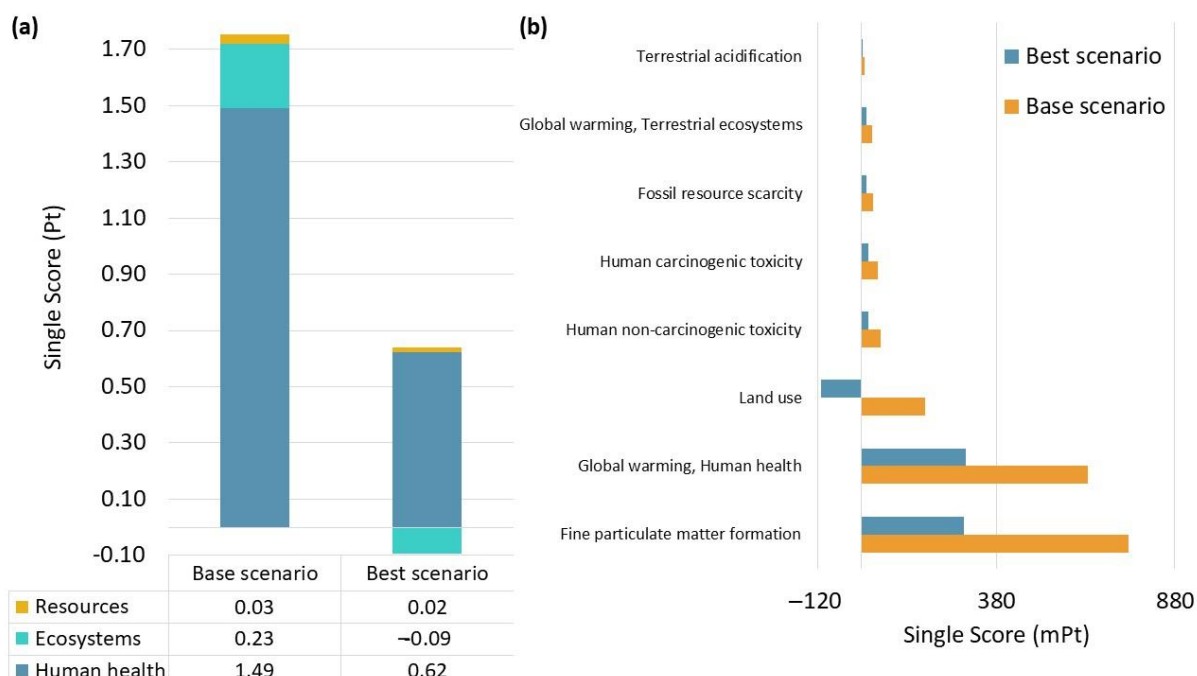

**Figure 9.** Comparison of the environmental impact of the base scenario compared to the best-case scenario ((**a**)—integrated results on environmental burdens of both scenarios, (**b**)—Scenarios compared between impact categories).

## 4. Limitations and Future Research Prospects

Despite the valuable insights gained from this study, there are certain limitations. Firstly, the study focused primarily on a theoretical analysis, and the scenarios were not validated through practical implementation in a pilot plant. Secondly, uncertainties associated with modelling and data inputs could impact the accuracy of the results. Furthermore, the study primarily examined a limited set of scenarios, and other factors influencing the environmental impact may not have been fully addressed.

The inclusion of uncertainties in modelling the environmental impact across various scenarios is a crucial aspect of conducting comprehensive research. By utilizing data obtained from operational pilot production plants, the potential for uncertainties can be significantly reduced. Nevertheless, even in cases where prospective studies involve high levels of uncertainty, they still provide valuable insights into enhancing the environmental performance of the technology during the design stage. Implementing improvements to the technology during the design stage not only yields cost advantages but also facilitates smoother full-scale implementation. This underscores the importance of integrating environmental considerations early in the design process, as it enables proactive measures to be taken to mitigate potential environmental impacts and maximize sustainability outcomes.

In terms of future research prospects, conducting experimental validation of the identified scenarios in pilot-scale operations would provide more accurate insights into their environmental performance. Additionally, exploring the potential impacts of varying process parameters and conditions, such as different fermentation conditions or alternative energy sources, could yield further optimization possibilities. Moreover, the integration of advanced genetic engineering techniques and innovative process designs could significantly enhance the sustainability of sophorolipid production. Lastly, a comprehensive LCA encompassing the entire supply chain, from raw material extraction to end-of-life, could offer a holistic view of the environmental implications and guide more informed decision-making.

## 5. Research Highlights and Conclusions

This study focused on the reduction of environmental impact assessment of the microbial fermentation of sophorolipids through the application of prospective LCA. The base scenario was compared with alternative scenarios of using waste substrate (waste cooking oil), increased or decreased sophorolipid titre, energy saving measures, and changes in bioreactor working volume.

The findings of this study contribute to the ongoing transition from crude oil and petrochemical refineries to sustainable biorefineries. Sophorolipid production, through the utilization of renewable resources, minimizing waste generation, and reducing electricity consumption, has potential in the transition to more sustainable production. By adopting more sustainable practices, the biorefinery industry can contribute to environmental improvements and promote the utilization of renewable bioresources.

Using LCA allowed identification of the environmental hotspots that had a significant environmental impact (>1%). Understanding these hotspots makes it possible to develop strategies for mitigating the environmental impacts. In this study, the identified hotspots were electricity consumption (85.1%) and choice of lipid source in fermentation substrate (14.2%). By understanding these hotspots, targeted strategies can be developed to mitigate their environmental impacts. Alternative scenarios were proposed and assessed to minimize the impacts.

Rapeseed oil used as a lipid source in fermentation substrate has a crucial role in the environmental performance of the fermentation process. In this study, the base scenario utilized RCO as the substrate. However, results indicated that WCO could be a more sustainable choice, as the use of WCO makes it possible to avoid 28% of the environmental impact that would be caused by cultivating rapeseed and producing RCO. The results of our study support the positive hypothesis that the use of WCO as a feedstock indeed resulted in a lower overall environmental impact compared to RCO. Another study that investigated the use of restaurant waste oil as a precursor for sophorolipid production suggested that this approach can reduce waste generation and promote the utilization of renewable resources [35]. Furthermore, by increasing the productivity of WCO utilization, as proposed in the study by Liepins et al. [32], it is possible to attain an even greater reduction of up to 50% in environmental impact.

The study also highlighted the impact of sophorolipid yield on the environmental performance of the fermentation process. To optimize the fermentation process for sophorolipid production, process improvements such as optimizing nutrient composition, fermentation conditions, and process control should be considered. Higher yields were found to result in more efficient resource utilization and reduced waste generation. This underscores the importance of ongoing research and development in genetic engineering, strain optimization, and fermentation process control to improve sophorolipid yield and further enhance sustainability. It is essential to highlight the importance of further investigating potential environmental impacts associated with scale-up of sophorolipid production. This can be achieved by utilizing available lab-scale plant data for first-generation sophorolipid production. By conducting a comprehensive analysis of these data, valuable insights into the scalability and sustainability of the sophorolipid production processes can be gained, enabling the development of more environmentally sustainable and efficient production methods.

Energy consumption was identified as a critical hotspot in the analysis, creating more than 80% of the total environmental impact, which aligns with the findings from Hu et al. [18]. The fermentation process requires energy for heating, mixing, and other operational requirements. The results emphasized the importance of implementing energy-efficient practices and process optimization to minimize the overall environmental impact. It confirms the hypothesis that optimizing the fermentation process led to a reduction in the total environmental impact of the production process. This finding suggests that biorefineries should prioritize energy-saving measures to reduce their carbon footprint and promote sustainability.

The compilation of the most environmentally favourable scenarios (best-case scenario), combining increased titres, optimized bioreactor working volumes, and varying electricity consumption, showed a 60% reduction in environmental impact when compared to the base scenario. Importantly, these findings align with our third hypothesis, which posited that the combination of these optimized scenarios would yield substantial environmental benefits. The reduction in environmental impact across various categories, particularly in land use, highlights the potential of utilizing WCO as a substrate for sophorolipid production. This approach not only reduces the overall environmental burden but also diminishes adverse effects on land use and ecosystems, underscoring its potential for sustainable production practices.

The inclusion of uncertainties in modelling the environmental impact across various scenarios is a crucial aspect of conducting comprehensive research. By utilizing data obtained from operational pilot production plants, the potential for uncertainties can be significantly reduced. Nevertheless, even in cases where prospective studies involve high levels of uncertainty, they still provide valuable insights into enhancing the environmental performance of the technology during the design stage. Implementing improvements to the technology during the design stage not only yields cost advantages but also facilitates smoother implementation. This underscores the importance of integrating environmental considerations early in the design process, as it enables proactive measures to be taken to mitigate potential environmental impacts and maximize sustainability outcomes.

**Supplementary Materials:** The following supporting information can be downloaded at: https://www.mdpi.com/article/10.3390/fermentation9090839/s1, Table S1: Inventory data for winter rapeseed production and rapeseed oil production adapted.

**Author Contributions:** Conceptualization, K.B., R.S., A.S. and E.D.; methodology, K.B.; investigation, K.B., A.S., K.D. and J.L.; writing—original draft preparation, K.B.; writing—review and editing, R.S., E.D., A.S. and J.L.; visualization, K.B. and K.D.; supervision, E.D.; project administration, E.D.; funding acquisition, E.D. All authors have read and agreed to the published version of the manuscript.

**Funding:** This research has been supported by the European Regional Development Fund within the project No. 1.1.1.1/19/A/047 "Sustainable Microbial Valorisation of Waste Lipids into Biosurfactants".

**Institutional Review Board Statement:** Not applicable.

**Informed Consent Statement:** Not applicable.

**Data Availability Statement:** The data presented in this study are available on request from the corresponding author.

**Conflicts of Interest:** The authors declare no conflict of interest.

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
