# Peer review of "Prospective Life Cycle Assessment of Microbial Sophorolipid Fermentation"

_fermentation, doi:10.3390/fermentation9090839_

Round 1

Reviewer 1 Report

The paper in general is good.

More citations should be made for the utilization of LCA and if it is possible for similar processes as here mentioned. 

Conclusions can be more punctual and related with results obtained. 

Reviewer 2 Report

I am grateful for the opportunity to review this paper.

I understand authors efforts behind this research, which is quite interesting, with great amount of results, with a suitable methodological approach and good quality of writing and scientific soundness. However, there are some parts that should be revised.

Comments:

(1) The paper demonstrates an adequate understanding of the relevant literature in the field and it cites an appropriate range of sources. The theme is interesting, well prepared as a research paper and fine presented.

(2) There is a clear presentation of the paper’s objective and a demonstration of it.

After going through all sections, I suggest the following:

(1)    Within the Introduction, there is a detailed explanation on the gap and contributions brought by this research and the methodology is also mentioned. Please, highlight the novelty of the proposed research (even if, while reading I could identify it by guess).

(2)    It really is a well-structured paper, however, while reading, I felt the need to see the   hypotheses your research is testing as related to your research aim.  You clearly explained what your intentions are (“This study focuses on improving the environmental performance of sophorolipid production using raw rapeseed cooking oil (RCO) as a lipid substrate in a lab-scale bioreactor. Prospective LCA methodology was used to quantify the environmental impacts of 83 the process (environmental hotspots) and to identify alternative production scenarios with lower negative impact on the environment. The primary aim of this article is to explore ways to improve the environmental sustainability of sophorolipid fermentation.”) but what are you expecting to find out? So, maybe in the Methodology section you can formulate some hypothesis and see in the end if you can support and validate them or if the findings reveal something else.

Good luck with future research!
